# Back to Clinical Training during the COVID-19 Pandemic: Perspective of Nursing Students

**DOI:** 10.3390/ijerph192114242

**Published:** 2022-10-31

**Authors:** Gregorio Jesús Alcalá-Albert, Eva García-Carpintero Blas, Cristina Gómez-Moreno, Carla González-Morón, Ana Sanz-Melero, Alejandra Sofía Robledillo-Mesa, Esperanza Vélez-Vélez

**Affiliations:** 1Nursing Department, School of Medicine, Alfonso X El Sabio University, 28691 Madrid, Spain; 2Fundación Jiménez Díaz School of Nursing, Autonomous University of Madrid, 28040 Madrid, Spain; 3Registered Nurse, Fundación Jiménez Díaz University Hospital, 28040 Madrid, Spain

**Keywords:** COVID-19, learning, nursing students, qualitative research, phenomenological design

## Abstract

Introduction: The COVID-19 pandemic has affected many areas of life, including the formation of nursing students. After the COVID-19 crisis, learning during clinical training created different challenges. Nursing schools are responsible for ensuring that structures are in place to facilitate coping in the changed clinical setting. This study aimed to analyze nursing students’ perceptions during clinical training while caring for COVID-19 patients. Material and methods: A qualitative phenomenological study that explored nursing students’ perceptions of learning in clinical settings with COVID-19 patients was performed. A total of 15 semi-structured face-to-face interviews were conducted with nursing students who carried out their clinical practices in COVID-19 units during February and April 2022. Results: Through content analysis, categorization, and the method of comparison constant, four categories emerged: feelings, challenges, coping methods, and clinical practices. The students had to learn to “work” with fear and uncertainty and self-manage the emotional burden using different coping techniques to deal with learning during their practices. Interacting with professors and clinical tutors during the clinical practice were positive experiences. Conclusions: This study constituted an opportunity to build new and adapted educational approaches for teachers to train nursing students to deal with their emotions and thoughts in future pandemic situations.

## 1. Introduction

The new coronavirus disease 2019 (COVID-19) was declared by the World Health Organization (WHO) as a public health emergency of international concern and a global pandemic [1]. This pandemic has affected many areas of life, including the academic education of nursing students, posing challenges in their clinical training [2]. Despite its continued expansion, nursing students are returning to learning and training on campus and in clinical settings. Distance education has replaced formal face-to-face education, and health sciences students have been the most affected by this situation [3].

Clinical practice is a critical yet complex and challenging component of professional development for nursing students [4]. Their professional performances are highly dependent on their clinical practices. It is known that some nursing students experienced anxiety during clinical training before the pandemic [5,6]. The academic and clinical stressors undergraduate nursing students perceive in their clinical learning environment have been the main objective of many studies [7,8]. Before the COVID-19 pandemic, some studies collected nursing students’ coping strategies to reduce stress at the beginning and during their clinical practices [9,10]. Therefore, the quality of clinical training is crucial for nursing education. Research published during the pandemic focused primarily on the anxiety and fears experienced by nurses in practice [11]. However, returning to learning in clinical settings during the pandemic is expected to generate a new source of anxiety, with feelings of insecurity among students [12]. Studies in different literatures have evaluated students’ stress and anxiety levels during the pandemic, regardless of clinical education [13,14,15]. Specifically, nursing students in clinical settings are under additional stressors and new challenges presented by COVID-19, such as fear of becoming infected and infecting their close family members [16], rapid changes in clinical settings [17], uncertainty about their skills, and even concerns about continuing their education [2,18]. There is evidence of their experiences during the first wave of the pandemic while attending academic courses [16] or clinical practices [19]. However, studies that raise the experiences in clinical training while caring for patients with coronavirus are scarcer [20,21].

Nursing schools are responsible for ensuring that structures are in place to facilitate coping and learning in the changed clinical setting [22,23]. They must play an active role in ensuring the personal safety of their students within the classroom and clinical settings. Nursing students’ use of problem-solving coping mechanisms has been documented in studies researching strategies to cope with academic and clinical stress [24,25]. In Spain, a bachelor’s degree in nursing, also known as Grado en Enfermería, is a four-year, eight-semester program corresponding to 240 ECTS (European Credit Transfer System). The program consists of theoretical and practical training. The practical training offered represents a high percentage of the total credits, exactly 81 ECTS, and is completed during the second, third, and fourth academic years. In this context, and on the light of previous research, exploring nursing students’ perceptions about their clinical practices in caring for patients with coronavirus is the main purpose of this study. The findings will allow nursing teachers to support and prepare students for the circumstances they will experience. In addition, the present study’s findings can help inform nursing curriculum developers on incorporating the necessary skills and resources to prepare future nurses for new pandemics.

## 2. Materials and Methods

### 2.1. Design

A qualitative study using Husserl’s descriptive phenomenological design [26] was used to explore nursing students’ experiences and coping strategies during clinical practices in units with COVID-19 patients. The purpose of phenomenological research is to describe particular phenomena, or the appearances of things, as lived experiences [27]. Since clinical training is a human experience, the use of phenomenological research is justified. Furthermore, narratives reveal the empirical qualities of a lived experience that cannot be separated from the context of the phenomenon [28].

### 2.2. Population and Sample

The study setting was a tertiary-level university hospital in Madrid, Spain. The participants were 2nd- or 3rd-year students from the FJD-UAM School of Nursing, who carried out clinical practices in at least one unit with COVID-19 patients in the 2020–2021 academic year. Those who had previous health work experience with these patients were excluded. A purposive and convenience sampling technique was used to recruit study participants who volunteered and agreed to participate. Data saturation was achieved in participant fifteen, and there were no dropouts.

The students performed their clinical practices for five weeks at the Fundación Jiménez Díaz University Hospital in morning or afternoon shifts together with a professional clinical care tutor. The units through which they rotated with COVID-19 patients were internal medicine, pulmonology, the intensive care unit (ICU), the intermediate care unit, and the emergency room. The students who decided to participate and met the inclusion criteria completed the information sheet and signed the informed consent before data collection.

### 2.3. Data Collection Instruments

The data were collected from February to April 2022 using in-depth semi-structured interviews to explore and interpret the participants’ perspectives on their lived experiences, providing all the possible information expressed in their own words and under their own subjectivity. A series of questions were asked that had previously been established in a script based on the previously consulted literature and the study’s objectives (Table 1). The sequence in formulating these questions could be modified depending on the subject interviewed.

Data collection was carried out by three researchers in a quiet and friendly room within the university facilities. In addition, field notes were collected by the researchers, adding information on the non-verbal language of the participants and recording essential information and incidents during their development [29]. The interviews were audio-recorded and lasted an average of 35–45 min. Confidentiality was guaranteed by consecutively numbering each interview and removing identification from the transcripts. Subsequently, transcriptions were emailed to the participants for verification and for participants to include or modify any of the information provided, if appropriate.

### 2.4. Data Analysis

A content analysis was performed through the categorization and constant comparisons method [30]. For this, the following phases described by authors such as Giorgi [31] were followed:Collection of information through semi-structured interviews.Careful reading after the literal transcription of the interviews.Decomposition to identify the units of meaning and relevant categories according to the objectives of the study and the information obtained.Organization and enumeration through a coding process.Interpretation, systematization, and data summarization to disseminate the results.

In the transcription phase, the audio and annotations of the interviews were literally transcribed for later interpretation. In this part of the analysis, alphanumeric codes were used to anonymize the participants (Table 2). All investigators participated in the transcription to become familiar with the data. Subsequently, data reduction was carried out, the texts were carefully read to select the most relevant information according to the study’s objectives. The Atlas ti-8 software (ATLAS.ti Scientific Software Development GmbH, Berlin, Germany) was used as computer support for this analysis [32].

### 2.5. Criteria of Rigor and Quality

The consolidated criteria for reporting qualitative research (COREQ) [33] were followed to ensure study quality. The rigor and scientific quality criteria of the qualitative studies proposed by Lincoln et al. [34] were also applied. The method to ensure credibility was the triangulation of data among researchers and validation of the findings by the participants. Reliability was achieved by maintaining consistency in the data collection process, specifically by using the same main questions in the interview guide across all participants. Finally, necessary information on the research context and methodology has been provided so that other researchers can replicate it, and the findings can be used to make informed decisions.

### 2.6. Ethical Considerations

This study was performed in line with the Declaration of Helsinki [35]. This study was approved by the Ethics Committee of the Fundación Jiménez Díaz University Hospital (registration PIC005-22_FJD). Students who decided to participate had to sign an informed consent, which was sent to the researchers by email.

Participants were informed that they could withdraw from the study at any time. They were asked if further clarification was needed about the study’s purpose and data collection process before the interviews started. The researchers also respected the rights of privacy, confidentiality, anonymity, and autonomy of the participants as pillars of justice and fair treatment of the subjects.

## 3. Results

Social-demographic features of the participants are shown in Table 2. The qualitative analysis of the collected data revealed four thematic categories: feelings, learning experiences, challenges, and coping methods. Table 3 shows the related categories and subcategories.

### 3.1. Feelings

#### 3.1.1. Fear

Almost all interviewees highlighted fear as the primary feeling before carrying out their internships in COVID-19 units and during them. They expressed fear and anxiety about clinical placements, now perceived as dangerous, due to the fear of becoming infected and a lack of knowledge and skills in caring for these patients.


*[…] well, a little bit of the feeling of fear…because after all, when you enter the ICU, it is like a parallel world, it’s like entering another part of the hospital, and it’s…a different type of patient, another type of everything, […]*
(E14)

Others highlighted the stress and concern generated by the fear of the possibility of contagion to their relatives, not so much for themselves.


*[…] I was scared because I was scared to get it and take it home, you know? Not only for me but my parents…because it is a disease […]*
(E6)


*[…] yes yes yes, eh…especially for that of my relatives because, you know, my father is at risk and, well, I was very afraid of that […]*
(E4)

#### 3.1.2. Anxiety

Anxiety was triggered not just for fear of contagion but also for previous psychological issues aggravated by COVID-19.


*[…] I have always had a lot of anxiety, especially this last year. Eh… and I suppose that with the COVID issue, it would get a little worse. And also being alert all the time like that, being aware and constantly thinking you can not get it, you can not take it home, you can not infect anyone […]*
(E2)


*[…] yah, yes… the situation made me anxious, yes…but was not related to COVID patients. I mean, yes, I was indeed a bit overwhelmed by the possibility of being or getting infected, but it wasn’t the patient, you know? […]*
(E8)

#### 3.1.3. Uncertainty

Some participants linked fear to a feeling of uncertainty due to a lack of experience and/or knowledge about caring for these patients with COVID-19.


*[…] The first thing I felt was a lot of respect, a lot of fear, and a lot of uncertainty because, on top of that I came from being a second-year student… And suddenly you see that you can be placed in an ICU, all COVID patients […]*
(E4)

#### 3.1.4. Stress

Some define clinical training as a stressful experience. Working in units full of infected patients and the inevitable constant contact caused stress and anxiety. Others expressed concern about getting infected as an exhausting situation or a constant insecurity.


*[…] well, stressful, yes. I think stressful is the word […] already at the end of it, and it was still stressful because no matter where you were, COVID was still around. At the very end of the covid period, there have already been fewer patients, so there has already been less accumulated stress […]*
(E11)

#### 3.1.5. Security

Most students agreed on a feeling of sufficient protection against COVID-19 with the material offered to them in the different units.


*[…] Yes, they always gave us all the needed material and even more because they didn’t want us…you know, I mean, they always had us under super surveillance, more than even themselves […]*
(E14)

However, other students reported a lack of agreement by the health professionals regarding the use of personal protective equipment (PPE) in the different COVID-19 units, which generated security for some and controversy for others, with some even feeling unprotected.


*[…] Me, personally, yes. And then I also saw the rest of the staff who were just as protected or less than me and… and I saw them calm, safe, and I said OK, it is OK […]*
(E4)

In addition, a general trend appeared during the clinical training; the perception of fear was transformed into a feeling of tranquility, trust, and protection as the students, during their clinical internships, gained more experience, especially among those working in the ICU.


*[…] as time goes by, I think you get used to it. I mean… now, for example, the ICU is not that bad, or I don’t see it like that. Probably there are more negatives, at least with those I was with, I don’t know. But what is true is that I wasn’t afraid of getting infected […]*
(E5)

### 3.2. Challenges

#### 3.2.1. Emotional Management

For the students, their own emotional management was their greatest challenge. Caring for these patients produced many feelings (fear, anxiety, frustration, uncertainty) that they had to work on and manage during their internships.


*[…] was… managing the fact of seeing a patient who… that is, the speed with which they died with which you changed patients, or with which, luckily, they were taken to a clean unit because I was… I mean, that… a little out of control […]*
(E1)

#### 3.2.2. Ignorance

It was challenging for the students to work with COVID-19 patients, as they had to deal with unfamiliar situations that were aggravated by their lack of knowledge about the aspects of caring for these patients.


*[…] Ehh…well (laughs), I have to think about it, the main challenge… well… I don’t know,… well yes, for me probably to understand everything that was around COVID. Yes, that was the most significant challenge, a whole different world, everything new, the medication, the positions of the patient […]*
(E12)

#### 3.2.3. Death

The students perceived COVID-19 patients’ deaths as very painful and with great sorrow, due to the bond generated between them from their long hospital stays.


*[…] at the end… they are so sick that if there is no other way out… It is like that, sadly, but there is no need to make a therapeutic obstinacy either. That makes me very sad […]*
(E5)


*[…] but it is true because it hurt me a little to think that I had been taking care of a person for so long and that in the end… it would end up like this, I don’t know, it gives you a lot to think about, you ask yourself a lot of things […]*
(E15)

In addition, this situation sometimes posed a challenge on a personal level.


*[…] I think fear of death… definitely the main one, yes, because I knew that I was going to a critical unit, that I was going to face it, and that… it had to happen to me at some point because in all the rotations I never saw anyone die, so I knew it was going to be the first time […]*
(E15)

Other students felt shocked because it was the first time they had witnessed either death or “therapeutic limitation”—end of treatment due to a poor prognosis—of a patient in the ICU.


*[…] Of course, it shocked me a bit, well… it was hard… truth, it shocked me a lot, scary, was a blow […]*
(E13)

#### 3.2.4. Management Caring Techniques

The care of patients with coronavirus required putting unknown caring techniques into practice. For some students, it was challenging to learn how invasive mechanical ventilation, tracheotomy cleaning, etc. worked.


*[…] And you know most of the techniques, you know? Hand hygiene, correct use of gloves, to put the underpad underneath, or how to mobilize patients, but with COVID patients, you feel more respect regarding their care…. not just because they are intubated, you know? and they are…. well… they are sick, some very very sick […]*
(E5)


*[…] and then well… handling techniques, which also gave me much respect and although I have learned a lot, you do them with a lot of fear […]*
(E15)

### 3.3. Coping Methods

#### 3.3.1. Family and Colleagues

Sharing the experiences with their loved ones helped them feel relief; it was their support tool.


*[…] mmm… yes, that is, I was supported by my friends and my family, my partner at the time and… yes, with them the truth is that they supported me, I… well, you also hear “wou, I admire you, in a Covid ICU!!-, you are…well…, not saving lives but what you do is meaningful […]*
(E4)

Some looked for that support among their colleagues, sharing experiences to cope with the situation:


*[…] in my colleagues because they had also been in units with COVID patients. Knowing their experiences in different COVID units helps a lot […]*
(E11)

Another coping strategy was disconnecting from the hospital once the shift was over, spending time and hanging around with friends.


*[…] especially with my boyfriend, he is extra!!. Uh… and… trying to escape from the hospital when I finish my shift, you know…. leaving everything out […]*
(E2)

#### 3.3.2. Clinical Training Tutors

Healthcare clinical tutors were the most important supports in the professional field. The students were reassured by discussing their concerns with them.


*[…] They explained things to me from the beginning, which was very comforting. In other words, all the time my tutors have never given me like… I don’t know…..anything to overwhelm me more […]*
(E5)


*[…] Well… I think the nurse I had in the ICU was fundamental. If she hadn’t been there, well, it would have scared me more, or even, i don’t know… anxiety […]*
(E12)

#### 3.3.3. Studying

Some participants used reading and studying as ways to manage their emotions during clinical practices and were able to feel calmer and unwinded.


*[…] yes, that is true because I have not done anything special, just read and brush up a little more to be more in control and calmer when working, you know?… but not much more […]*
(E15)

#### 3.3.4. Psychological Therapy

A student needed psychological therapy to deal with anxiety. Other students commented on going to therapy, but it was not because of clinical internships. However, this resource helped them better manage their anxiety levels during clinical training.


*[…] different situations that happened in my family also affected me, not because of COVID itself, but because of mental health due to COVID, and that affected me, and I’ve been pretty bad for two years… and… this year I started with psychologists […]*
(E10)

### 3.4. Clinical Practices

#### 3.4.1. Learning Richness

Several students agreed that clinical practice provided them with a series of necessary knowledge and techniques that were unknown to them concerning the treatment of patients with COVID-19 and everything the disease entailed.


*[…] but, although… some patients have been very ill, I must say I have learned many things regarding the care of a respiratory patient. And I think I would have never learned if it hadn’t been so. I mean, on the one hand, I am thankful for having this placement at this time […]*
(E15)

#### 3.4.2. Multidisciplinary Care

Participants emphasized that taking care of these patients involved a wide variety of techniques, medication, and healthcare activities to achieve their improvement, with the participation and support of a multidisciplinary team.


*[…] things were being tested […] there were clinical trials, one medication was tried, then another […] It was very dynamic […]*
(E1)

#### 3.4.3. Gratification

Participating in the care of these patients, seeing how they evolved and improved, and the knowledge learned made the participants define the experience of their clinical practices as a very rewarding and satisfying situation.


*[…] How would I say it […] as…in general, as one… eh, pleasant experience, you know? it can be very gratifying to see when a person who is very sick and suffering recover […]*
(E3)


*[…] I realized how much more satisfying it was to be treating a patient that you say: “I see him objectively bad, very sick, and I know that maybe what I am giving him is not going to cure him completely, but it is going to make him feels better” […]*
(E7)

## 4. Discussion

Clinical practice is a major component of nursing education in which meaningful learning takes place. Nursing students view clinical training as a source of concern and stress, even under normal circumstances. Their most significant sources of stress are lack of competence and fear of causing harm to patients. [36]. Their return to the clinical field to carry out their clinical training in COVID-19 units confronted them with a new unknown scenario, full of uncertainties and fears influenced by the environment and its different contexts. This study’s findings indicate that most students felt fear, anxiety, stress, and uncertainty related to the COVID-19 infection and the lack of knowledge about taking care of these patients. Similar results have been reported in studies on returning to clinical learning after the COVID-19 outbreak [12,37,38]. However, there is also evidence showing that stress experienced by students during their clinical training was not different from stress experienced before the pandemic; furthermore, fear of COVID-19 did not affect the level of perceived stress in the clinical education setting [39]. John and Al-Sawar found that almost all nursing students experienced moderate to severe stress levels during their clinical training [40]. 

Our participants reflected that a key fear and source of anxiety was protecting family and self-protection from contagion. These perceptions are consistent with other studies showing that the three most important goals in science students’ future professional performance are family security, a sense of achievement, and happiness. [37,41]. Nonetheless, these concerns did not prevent students from assuming clinical training with full dedication and motivation, feeling gratification, and expanding their knowledge; these findings have also been reported in other countries [42,43].

Nursing students recognized the challenges encountered during their clinical practices. As in other studies, emotional management was the greatest challenge faced during clinical internships with COVID-19 patients [44]. They also expressed sadness and shock at witnessing and confronting death for the first time and at the therapeutic limitations of some patients, something reflected in other studies [45,46].

As before the pandemic, students used coping strategies to deal with the impacts that COVID-19 had on their psychological well-being. A study published before the pandemic found that nursing students’ most common coping behaviors were transference, optimism, and problem-solving. At the same time, the least used was avoidance [47]. However, a recent study on student coping strategies during COVID-19 found that approximately 35% experienced some anxiety and used four coping strategies: seeking social support, avoidance/ acceptance, mental disengagement (disconnection from the negative situation and engagement in other activities), and humanitarian coping (reaching out to others and attempting to helping them) [48]. 

In the present study, family and friends were shown to have a supportive and influential role in providing a protective environment for students. Consistent with other studies, students’ coping resources increased when the family was perceived to be supportive, facilitating adaptive coping [37]. In addition, the participants reflected that the role of clinical training tutors in dealing with these situations was key to having a positive experience in their clinical practices. These results coincide with other studies that reflect the critical roles of universities and nursing professors in emotionally supporting students and guaranteeing their safety in the classroom and clinical settings [16].

### 4.1. Strengths and Limitations

The specific environment of the study and the qualitative method applied do not allow generalizations. However, the results can make an exciting contribution to understanding similar experiences.

Future studies should analyze students’ perceptions during their internship in other settings such as nursing homes, health care centers, etc.

Further quantitative studies should measure and analyze the stress suffered by students as continuous companion during the internship, before and during the pandemic, and their relations with the resilience levels of the students.

### 4.2. Implication for Practice and Research 

Implementing support measures for students in the transition to situations of uncertainty, such as the pandemic, is essential. Integrating a specific module of emotion management into the basic training program of nursing students to promote the psychological health of nursing students is a must. Emotional management in general is a big issue in practical nurse training; with and without a pandemic crisis, stress is always there, as this study has shown. 

The results of this study leave open a line of research. Resilience is a significant concept of positive psychology. It refers to the dynamic process and positive adaptation to bitter and unpleasant experiences [49]. Resilience and its components, such as self-confidence, optimism, belief in individual capabilities, and acceptance, act as barriers to stressful situations and increase human strength and tolerance [50]. Further studies on the correlation between anxiety, stress, and resilience levels in nursing students are suggested.

## 5. Conclusions

This study comprehensively analyzes students’ feelings, perceived challenges, and coping strategies during their clinical practices in COVID-19 units. Identifying their lived experiences in clinical training and finding solutions to the problems detected is essential. 

Our findings indicated that nursing students experienced significant anxiety related to COVID-19 infection upon returning to on-campus learning. The students had to learn to overcome fear and uncertainty and self-manage the emotional burden using different coping techniques to deal with learning during their practices. Interacting with professors and clinical tutors during the clinical practice was a positive experience. This study constitutes an opportunity for teachers to build new and adapted educational approaches to train nursing students to deal with their emotions and thoughts in future pandemic situations.

## Figures and Tables

**Table 1 ijerph-19-14242-t001:** Topics and questions.

Topics	Questions
Learning	How would you describe your clinical internship experience caring for COVID patients? Do you find any difference in having carried out clinical practices in units without COVID patients and later treating COVID patients? Which? How has rotating through COVID units influenced your internship?
Covid	Did you get infected with the virus during your clinical practices? Did you fear the contagion and the health of your relatives? Do you think you felt protected treating patients with COVID-19?
Challenges	What do you think has been the main challenge or difficulty that you have had to face during this clinical internship period? What feelings did you have during the care of these patients?
Coping	How did you manage your feelings and fears while providing nursing care to COVID patients? Have you relied on someone / used any tool for it?

**Table 2 ijerph-19-14242-t002:** Participants’ demographic data.

Participants	Grade	Gender	Age	Unit	Infected with COVID	Live with Family Members
E1	3rd	Woman	23	ICU	Yes	No
E2	3rd	Woman	21	Internal Medicine	No	Yes
E3	3rd	Men	21	ICU	No	Yes
E4	3rd	Woman	21	ICU	No	Yes
E5	3rd	Woman	21	Emergency Room	Yes	Yes
E6	3rd	Woman	21	Emergency Room	Yes	No
E7	3rd	Woman	21	Internal Medicine	No	Yes
E8	3rd	Woman	21	ICU	No	Yes
E9	3rd	Woman	21	ICU	Yes	No
E10	3rd	Woman	21	ICU	No	Yes
E11	3rd	Woman	20	Internal Medicine	No	Yes
E12	3rd	Woman	25	Pneumology Unit	No	Yes
E13	2nd	Woman	23	Internal Medicine	Yes	Yes
E14	3rd	Woman	20	ICU	No	Yes
E15	3rd	Woman	23	ICU	Yes	Yes

**Table 3 ijerph-19-14242-t003:** Categories and subcategories matrix.

Categories	Subcategories
Feelings	Fear Anxiety Uncertainty Stress Motivation Frustration Security
Challenges	Emotional management Ignorance Death Management caring techniques
Coping methods	Family and colleagues Clinical training tutors Studying Psychological therapy
Clinical practices	Learning richness Multidisciplinary care Gratification

## Data Availability

Raw data from the interview transcripts supporting the findings of this study are available from the corresponding author upon reasonable request. These data are not publicly available due to them containing information that could compromise the privacy of research participants.

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
