# Peer review of "Back to Clinical Training during the COVID-19 Pandemic: Perspective of Nursing Students"

_ijerph, 2022, doi:10.3390/ijerph192114242_

Round 1
Reviewer 1 Report
Introduction
As it is a qualitative study it is essential to introduce the social context of the research. Where was it done (country)? What is the specificity of nursing studies there (e.g. how long do studies last)? When did the clinical setting begin in nursing studies?
At the end of the introduction, there should be a straightforward research question.
Methods
Does the research have the Ethical Commission agreement?
Does the transcription publicly available?
Who conducted the interviews?
Results
I think the first line is quantitative and unnecessary, just leave the introduction that social-demographic features of respondents can be found in table 2.
Discussion
It is a very short part and needs to be extended. You showed very interesting results but did not discuss them well.
I did not find the answer to the questions: What future research should be made? What are the limitations of the study?
Conclusion
There should be a clear answer to your research question.
Author Response
Thank you so much for your in-depth revision and your comments and corrections. We are sure they have enhanced the clarity of the manuscript.
Thank you for your comments, which allow us to improve the quality of the manuscript.
As it is a qualitative study, it is essential to introduce the social context of the research. Where was it done (country)? What is the specificity of nursing studies there (e.g. how long do studies last)? When did the clinical setting begin in nursing studies?
Thank you for your suggestions: The country and the specificity of nursing studies and clinical practice in this country have been included.
At the end of the introduction, there should be a straightforward research question.
As suggested, the purpose of the study has been included in the last paragraph of the introduction.
Methods
Does the research have the Ethical Commission agreement?
Yes, thank you. We have included the ethical considerations epigraph (pp 5, lines 165 to 172), where the approval by the Research Ethics Committee is mentioned. In addition, the Institutional Review Board Statement is completed at the end of the article.
Does the transcription publicly available?
As mentioned in the "Data Availability Statement,": Raw data from the interview transcripts supporting the findings are not publicly available due to containing information that could compromise the privacy of research participants.
As an aspect of ethics in Qualitative research, the authors got the interview transcript approval from the participants. Once transcriptions were finished, they were passed to the participant to get their approval.
Who conducted the interviews?
Three of the researchers. The entire research team carried out the analysis, as explained in the data analysis section (pp.4, line 114 and 138).
Results
I think the first line is quantitative and unnecessary, just leave the introduction that social-demographic features of respondents can be found in table 2.
Thank you. Changed as suggested
Discussion
It is a very short part and needs to be extended. You showed very interesting results but did not discuss them well. I did not find the answer to the questions: What future research should be made? What are the limitations of the study?
Thank you for your suggestions. The discussion has been thoroughly reviewed. Sub-epigraph "strengths and limitation" and "implication for practice and research" have been added at the end of the discussion section.
Conclusion
There should be a clear answer to your research question.
Thank you. Added as suggested. The main findings are pointed out.
All changes have been marked in yellow in the text.
Reviewer 2 Report
In general, this paper addresses a very important and interesting topic. The abstract is very clear and contains all important information. In order to better understand the results, it would be helpful to have an insight into the results of previous studies before the COVID-19 pandemic.
Perhaps the authors already have a suggestion on how to better prepare nurse students for the challenges of working life. Strategies to improve resilience could be a possible approach to a solution.
As far as I have understood correctly, there is no difference in the psychological stress before or during the pandemic - this should be highlighted (apart from the fear of contracting with Covid) - or it should be explicitly pointed out where there were concrete differences.
Introduction:
Very compact. In order to get a better understanding of the emotional challenge (stress and anxiety) of the nursing students in general, it would be helpful to briefly present the results of previous studies before the pandemic as well as meanwhile.
Materials and Methods:
Ad Table 2:
2 questions remain open for me:
1. Did the authors include also the male participant?
2. Related to the column “Covid”: does this mean that the got infected with Covid-19 during their clinical training?
Discussion:
The authors mentioned that stress is a continuous companion during the internship and that the stress level has not differed before and during the pandemic. It would be generally interesting how these results are connected with the resilience level of the students (see for example Mohammadi et al., 2022).
As far as I can see in the literature, emotional management in general is a big issue in the nurse practical training, and cannot be related exclusively to the COVID situation.
Conclusion:
Even though the relationship between resilience and mental health in nurse students has not yet been sufficiently researched, this might be an additional possibility to think about in order to prepare nurses for the stressful daily routine in the best possible way.
Author Response
Thank you so much for your in-depth revision and your comments and corrections. We are sure they have enhanced the clarity of the manuscript.
Thank you for your comments, which, we are pretty sure, have allowed us to improve the quality of the manuscript.
In general, this paper addresses a very important and interesting topic. The abstract is very clear and contains all important information. In order to better understand the results, it would be helpful to have an insight into the results of previous studies before the COVID-19 pandemic.
Thank you so much. The article has been expanded with more sources and bibliographical references, including studies prior to the pandemic.
Perhaps the authors already have a suggestion on how to better prepare nurse students for the challenges of working life. Strategies to improve resilience could be a possible approach to a solution.
Yes, thank you, we agree. In fact, currently, we are analyzing this potent concept using a different study design.
As far as I have understood correctly, there is no difference in the psychological stress before or during the pandemic - this should be highlighted (apart from the fear of contracting with Covid) - or it should be explicitly pointed out where there were concrete differences.
Thank you, this is a good point. We have not measured the stress level, as this was not the goal of this study. However, stress is there, with and without the pandemic, as different studies point it out. Studying the differences, if existing, could be the subject of another qualitative study. The intensity or difference in intensity could be studied with a quantitative approach.
Introduction:
Very compact. In order to get a better understanding of the emotional challenge (stress and anxiety) of the nursing students in general, it would be helpful to briefly present the results of previous studies before the pandemic as well as meanwhile.
Thank you. We expanded this section as suggested with more studies prior to the pandemic.
Materials and Methods:
Ad Table 2:
2 questions remain open for me:
- Did the authors include also the male participant?
Yes, all participants of the study are included in table 2.
- Related to the column “Covid”: does this mean that the got infected with Covid-19 during their clinical training?
Yes, thank you. For more clarity we have improve that tag, including “infected with COVID”. Students got covid during their clinical training.
Discussion:
The authors mentioned that stress is a continuous companion during the internship and that the stress level has not differed before and during the pandemic. It would be generally interesting how these results are connected with the resilience level of the students (see for example Mohammadi et al., 2022).
Thank you for your comments. Your suggestions, very well brought up, have been taken into account, and we have included them.
As far as I can see in the literature, emotional management in general is a big issue in the nurse practical training, and cannot be related exclusively to the COVID situation.
Yes, we agree; thank you. This is something we specify in "implication for practice"
Conclusion:
Even though the relationship between resilience and mental health in nurse students has not yet been sufficiently researched, this might be an additional possibility to think about in order to prepare nurses for the stressful daily routine in the best possible way.
Thank you so much. We totally agree. This is something we specify in implication for practice and conclusion.
All changes have been marked in yellow in the text.
Round 2
Reviewer 1 Report
N/A
Author Response
Thank you for your comments and suggestions. The misspelling has been corrected.
Besides, a native English speaker reviewed the manuscript a second time to improve style, some sentences were rewritten to minimize wordiness, others to improve clarity, and some words changed for consistency.
All changes are marked in red in the manuscript. In addition, the details added here:
Pg 1, line 23. Changed “by” for “for”
Pg 2. Line 59. Changed “how to incorporate” for “incorporating” (minimizing wordiness)
Pg 2, lines 74-76. Sentence rewrite for clarity
Pg 2, in the table. Erased “for”, unnecessary preposition
Pg 5, in the table. Changed “twenty” for “20” (consistency)
Pa 5, lines 151 and 154. Replaced “rigor” with the British English spelling “rigour.”
Pg 9, line 301. Rewritten to minimize wordiness
Pg 10, line 327- Rewritten for clarity
Pg 10, line 332. Erased unnecessary article
Pa 10, lines 353. Changed preposition “on” for “about” (grammar correctness)
Pg 11, line 374. Replaced “behavior” with the British English spelling “behaviour”
Pg 11, line 400: Changed “the implementation of” for “Implementing” (conciseness)
Pg11, line 401. Changed “Integrate” for “integrating” (Grammar correctness)
Pg11, line 405: Changed “have” for “has” (grammar correctness)